# Effect of Acrylonitrile Butadiene Styrene (ABS) Secondary Microplastics on the Demography of *Moina macrocopa* (Cladocera)

**DOI:** 10.3390/biology14050555

**Published:** 2025-05-16

**Authors:** Diana Laura Manríquez-Guzmán, Diego de Jesús Chaparro-Herrera, Pedro Ramírez-García, Cesar Alejandro Zamora-Barrios

**Affiliations:** 1Laboratorio de Microecología Ambiental UIICSE-CyMA, National Autonomous University of Mexico, Campus Iztacala, Av. de Los Barrios No. 1, Los Reyes, Tlalnepantla C.P. 54090, State of Mexico, Mexico; dlmanriquez@ciencias.unam.mx (D.L.M.-G.); micro@unam.mx (P.R.-G.); 2Laboratory of Water Pollutants Removal Processes, Division of Research and Postgraduate Studies, National Autonomous University of Mexico, Campus Iztacala, Av. de Los Barrios No. 1, Los Reyes, Tlalnepantla C.P. 54090, State of Mexico, Mexico

**Keywords:** microplastics, *Moina macrocopa*, ABS-MPs

## Abstract

This study investigates the effects of secondary microplastics, specifically acrylonitrile butadiene styrene, on the biology of *Moina macrocopa*, a freshwater zooplankton species. The research focuses on how these microplastics influence key demographic indicators such as survival, mortality, life expectancy, and fecundity. *M. macrocopa* were exposed to three chronic concentrations of microplastics: 5, 10, and 20 mg L^−1^.

## 1. Introduction

Since their first production in the 1950s, plastic materials have been considered an innovative product utilized in daily life. However, their excessive consumption, inadequate recycling, and difficulty in removal have led to environmental problems due to the high concentrations recorded [1,2]. Plastic particles smaller than 5 mm are classified as microplastics (MPs) and are almost imperceptible due to their size [3,4]. Primary MPs are manufactured industrially, such as microspheres for cosmetics, toothpaste, and facial scrubs. On the other hand, secondary MPs are formed by fragmentation or degradation of larger plastic items [5]. Nowadays, MPs are considered an emerging pollutant found in terrestrial and aquatic ecosystems, including coral reefs, Antarctic deep waters, and Arctic glaciers [6,7,8]. Research on microplastic pollution in aquatic environments is focused on marine habitats. However, freshwater ecosystems are also vulnerable due to direct anthropogenic influence, with plastic concentrations that could be comparable to, or even exceed, those observed in marine systems [9,10,11].

MPs access freshwater environments through different pathways, including treated water effluent, wind, stormwater runoff, surface runoff, and activities such as tourism [12,13]. Once MPs reach waterbodies, they negatively impact aquatic organisms when ingested voluntarily or involuntarily, disrupting the digestive tract and causing damage at all levels of biological organization [14,15]. The risk associated with MPs is also related to additives present in plastic materials, such as plasticizers, antioxidants, heat stabilizers, or slip agents. Additionally, MPs act as vectors for pollutants through adsorption and can host pathogens that adhere to the plastisphere [16,17].

Zooplanktonic organisms have an essential function in aquatic food webs as primary consumers [18]. Despite their importance and vulnerability, studies on their interaction with plastic particles are less common than those focused on mollusks and fish [19,20]. Zooplankton consume MPs through non-selective filtration or by mistaking them for prey, resulting in false satiation. This reduces their survival by interfering with the energetic budget associated with the presence of plastic particles and decreasing their natural ingestion rates, as observed in rotifers, copepods, and cladocerans [21,22,23]. The damage caused by MPs is related to factors such as size, shape, type, and concentration [24,25].

Most studies examining the impact of MPs on zooplankton have focused on standard polystyrene (PS) and polyethylene (PE) materials with spherical shapes and uniform sizes. However, in freshwater environments, most microplastics consist of irregular fragments and fibers of varying sizes [19]. In addition, other plastics commonly found in aquatic ecosystems include polyvinyl chloride (PVC), polyethylene terephthalate (PET), and acrylonitrile butadiene styrene (ABS), which are often overlooked [19,26]. ABS plastics are produced in large quantities, are highly toxic, and are difficult to biodegrade. Despite their persistence, they have not been widely utilized in aquatic toxicity studies [27,28].

The genus *Moina* is a common cladoceran in tropical freshwater bodies, particularly in polluted waters. They exhibit high densities throughout the year, making them an ideal organism to examine sensitivity to environmental stressors [29]. Their transparent body allows direct observation of contaminants through their gut, and their short lifespan and parthenogenetic reproduction make them suitable for both short-term (acute) and long-term (chronic) assessments. In addition, they are recognized as bioindicators, which increases their importance in assessing the effects of toxic substances, including microplastics [25]. This research aims to evaluate the effects of secondary ABS-MPs with irregular shapes and several sizes on demographic indicators such as survival, mortality, life expectancy, and fecundity in *Moina macrocopa*.

## 2. Materials and Methods

### 2.1. Moina macrocopa and Chlorella vulgaris Maintenance

The parthenogenetic populations of the cladoceran *Moina macrocopa americana* used to establish the stock cultures were obtained from the culture collection of the Aquatic Zoology Laboratory of UNAM, Mexico. These populations were isolated from the Ramsar site Lake Xochimilco and have been maintained for more than five years under controlled laboratory conditions. Cultures were established in EPA medium (Environmental Protection Agency medium, moderately hard water), consisting of 1.9 g NaHCO_3_, 1.2 g CaSO_4_, 1.2 g MgSO_4_, and 0.04 g KCl dissolved in 20 L of distilled water [30]. Stocks were maintained in 1 L glass beakers at room temperature with 12:12 h photoperiods. Their medium was renewed every three days, and the microcrustaceans were fed with the green microalga *Chlorella vulgaris* as an exclusive diet.

Axenic cultures of the microalga *C. vulgaris* (strain CL-V-3 CICESE, MX) were grown using a standardized Bold medium [30] supplemented with 0.5 g NaHCO_3_ every three days as an additional carbon source. Cultures were established in 2 L graduated Erlenmeyer flasks with 1.5 L of sterile medium. Initial microalgal inoculation was initiated with 1 × 10^5^ cells mL^−1^. The algae were grown in a batch regime at a temperature of 25 ± 2 °C under a 16:8 h photoperiodic cycle and an irradiance of 150 μmol m^−2^ s^−1^. Moderate bubbling aeration was used to avoid cell sedimentation. *C. vulgaris* cells were decanted, the supernatant was removed, and the cells were rinsed and subsequently resuspended in an EPA medium. The algal cell density was quantified as the number of cells per mL using a hemocytometer (Neubauer chamber).

### 2.2. Preparation of ABS Microplastics

The small polymeric pieces were collected from an industrial ABS plastic manufacturing plant located in Mexico City. The synthetic material was mechanically shredded several times to create secondary ABS-MPs of less than 5 mm. The shredded plastics were then sieved to obtain ABS-MPs with sizes between 10 and 150 µm. The size of the ABS-MPs was verified microscopically. Dust and other contaminants were removed from the surface of the ABS-MPs by washing them with ethanol, rinsing with distilled water, and drying at 45 °C. The ABS-MPs were then resuspended in distilled water, and the number of plastic particles was obtained using a Sedgewick–Rafter chamber. Resuspended ABS-MPs were diluted in an EPA medium to prepare a stock solution from which working concentrations used in the bioassays were obtained. As MPs tend to adhere to the forming plastic beds, the stock solution was sonicated for 3 min at 20 KHz before use.

### 2.3. Ingestion of Moina macrocopa to ABS-MPs

To analyze the ingestion of *M. macrocopa*, neonates of the third brood that were less than 6 h old were used. The microcrustaceans were fed ad libitum with *C. vulgaris* for 24 h prior to the experiments. Subsequently, ten neonates were randomly transferred to 50 mL borosilicate vials containing 40 mL of EPA medium and subjected to a one-hour starvation period. Neonates were fed for 30 min with an initial concentration of 0.5 × 10^6^ cells of *C. vulgaris* mL^−1^. Ingestion rates were evaluated under two conditions: *C. vulgaris* alone and combined with ABS-MPs at concentrations of 5, 10, and 20 mg L^−1^. Each test was performed in quadruplicate at room temperature. At the end of the filtration period, neonates were removed, and unconsumed cells were immediately fixed with 3% formaldehyde to ensure accurate quantification. The concentration of the cells was measured using a Neubauer chamber, and the ingestion rate was calculated by finding the difference between the initial and final algal densities.

### 2.4. Exposure of Moina macrocopa to ABS-MPs

The chronic experiments were performed as follows: 10 *M. macrocopa* neonates (less than 24 h born) were placed in 50 mL transparent containers with 40 mL of EPA medium filtered with a cellulose filter (Millipore^®^ St. Louis, MO, USA) to avoid contamination. Each group of neonates was exposed to one of three concentrations of ABS-MPs (5, 10, and 20 mg L^−1^). All experiments were performed in quadruplicate, with each replicate containing *C. vulgaris* at a concentration of 0.5 × 10⁶ cells mL^−1^ as the sole food source. A control group without plastic particles subjected to the same diet as the toxic treatments was included (Figure 1). The vessels were covered with aluminum foil to prevent external contamination by airborne particles. The experiment was maintained under a 12:12 h light/dark cycle at room temperature (22 to 24 °C, confirmed with data recording equipment). Every 24 h, the EPA medium was replaced with *C. vulgaris* and ABS-MPs. Using a stereo microscope (Nikon SMZ800), the number of live and dead cladocerans was determined on a daily basis. Neonates were removed to exclusively monitor the initial cohort. With the data obtained, the demographic curves of survival, mortality, and fecundity were plotted for each treatment.

In addition, the demographic variables of average lifespan, net reproduction, generation time, and population growth were calculated [31]. In this instance, *x* represents age; *l_x_* is the proportion of individuals surviving to age *x*; *m_x_* denotes the proportion of females born during time interval *x*; *e_x_* corresponds to the average life expectancy of organisms alive at the beginning of the lifespan; *T_x_* is the total number of individuals alive on average at age 0; and *n_x_* indicates the initial population size at age 0. The constant *e* in the population rate equation is equal to 2.71828; *r* represents the intrinsic rate of population increase.

Average lifespan$$e_{x}=\frac{T_{x}}{n_{x}}$$

Net reproductive rate$$=\overset{\infty }{\underset{0}{\sum }}l_{x}m_{x}$$

Generation time$$T=\frac{\sum l_{x}m_{x}x}{R_{0}}$$

Rate of population increase$$\overset{n}{\underset{x=w}{\sum }}e^{-rx}l_{x}m_{x}=1$$

### 2.5. Statistical Analysis

Statistical analyses were performed by analysis of variance (ANOVA) followed by Tukey’s test to identify significant effects. Normality was assessed using the Kolmogorov–Smirnov test and homogeneity of variances was evaluated using Levene’s test. All analyses were performed using SigmaPlot^®^ 11 software (Systat Software, Inc., San Jose, CA, USA).

## 3. Results

### 3.1. Cell Ingestion of Moina macrocopa

The presence of microplastics altered the number of cells ingested during the filtration process of *M. macrocopa*. The general trend showed that as the concentration of ABS-MPs increased, the number of food cells ingested decreased significantly (Figure 2). In the control treatment, the cladocerans consumed 625 cells of *C. vulgaris* ind^−1^ min^−1^. In contrast, at 20 mg L^−1^, an 85% reduction (94 cells ind^−1^ min^−1^) was observed. Neonates exposed to 5 mg L^−1^ ABS-MPs consumed 341 cells ind^−1^ min^−1^, which corresponded to approximately half the amount observed in the treatment without ABS-MPs. This indicates that the presence of ABS-MP influences filtration, even at low concentrations. All treatments differed significantly from the control, while no significant differences were observed between the 10 mg L^−1^ and 20 mg L^−1^ treatments (Table 1).

### 3.2. Survivorship

Survival of cladocerans exposed to microparticles decreased compared to the control treatment, which showed a mean survival of 16 days. In contrast, the ABS-MP treatments resulted in survivorship ranging from 11 to 13 days (Figure 3). The Kaplan–Meier survival analysis indicated a significant difference in survival curves (*p* = 0.038). However, the Holm–Sidak *post hoc* test (*p* < 0.05) suggested that although there was an overall trend towards reduced survival, the variations between groups were not large enough to be statistically significant. Survival curves for controls indicated that cladocerans died mainly in the last days of their life cycle, corresponding to a type I curve. In treatments with the presence of ABS-MPs, *M. macrocopa* showed a decrease in survival, with a reduction of ~39% in the first four days. By day 10, when the control population reached a mean survival rate of 0.5 (lx), toxic treatments showed a 70% decrease in survivorship.

### 3.3. Mortality Rate

The particulate-free treatment (control) showed low mortality rates during the first 11 days. After this period, the population began experiencing senescence, resulting in a continuous increase in mortality. In the particle concentration treatment of 5 mg L^−1^, the lowest mortality rate (qx) was observed from days four to eight. In the 10 mg L^−1^ treatment, the lowest mortality rate occurred between days two and seven, while in the highest concentration treatment of 20 mg L^−1^, mortality rates remained low between days two and six of life (Figure 4).

### 3.4. Fecundity

Fecundity curves of *M. macrocopa*, evaluated by age-specific analysis, showed that all treatments reached reproductive maturity at 72 h (day three). In the control treatment, this variable remained constant for 17 days. On the contrary, in the ABS-MP treatments, particularly in the concentrations of 10 and 20 mg L^−1^, the fecundity was inversely proportional to concentration, showing a 37.7% reduction compared to the control (Figure 5). Nevertheless, the 10 mg L^−1^ treatment exhibited the highest reproductive rates, with a peak production of 22.1 neonates per female per day, representing a 35.8% increase in reproductive effort compared to the control. Fluctuations in fecundity seen in the ABS-MP treatments were most likely due to the toxic effects of MPs or changes in energy cost. Throughout the experiment, MPs were observed stored in the thoracic appendages of cladocerans, in some cases blocking the intestine and in others adhering to the carapace (Figure 6).

### 3.5. Life Table Parameters

The life expectancy (e_x_) of *M. macrocopa* in the control group was 9.48 ± 0.09 days. On the other hand, in the ABS-MP treatments, it decreased significantly to 6.23 ± 0.74, 6.98 ± 0.31, and 7.03 ± 0.63 days for 5, 10, and 20 mg L^−1^, respectively. A one-way ANOVA confirmed significant differences (*p* < 0.01), indicating that the presence of MPs had a negative effect on this demographic variable (Table 2). However, MP concentration did not result in statistically significant differences. Tukey’s post hoc test (*p* < 0.05) further confirmed that all ABS-MP treatments resulted in a comparable reduction in life expectancy, suggesting that even the lowest concentration (5 mg L^−1^) had a negative impact on the survival of *M. macrocopa*, reducing it by 34.3%.

The net reproduction rate (R_0_) showed a significant decrease in response to MP exposure. In the control group, the mean value was 55.93 ± 5.89, while all treated groups showed lower means: 30.75 ± 13.39 (5 mg L^−1^), 39.50 ± 8.12 (10 mg L^−1^), and 32.90 ± 6.35 (20 mg L^−1^) (Figure 7). The greatest variability in responses was observed in the 5 mg L^−1^ treatment, suggesting inconsistent responses amongst individuals. Remarkably, the 10 mg L^−1^ group presented an intermediate effect, while the 20 mg L^−1^ treatment did not show a clear dose-dependent trend, as its mean value was comparable to that of 5 mg L^−1^. The 5 mg L^−1^ concentration caused a reduction of 25 offspring per female, representing approximately 45% fewer compared to the control group (*p* < 0.05, Tukey’s test). Nevertheless, the same was observed at the medium and high concentrations, which were significantly different from the control but not from the 5 mg L^−1^ treatment (*p* > 0.05, Tukey’s test). These results indicate that ABS-MPs affect reproductivity, but the response does not follow a strictly linear pattern with increasing concentrations.

Cladocerans exposed to MPs showed a generation time of 6.28 ± 0.68 days for 5 mg L^−1^, 6.36 ± 0.70 days for 10 mg L^−1^, and 6.15 ± 1.16 days for 20 mg L^−1^ (Figure 7). The significant decrease in generation time was due to fragmented plastics exposure (*p* < 0.05). In the control group, the mean was 8.33 ± 0.77 days, which is 24–26% longer. Although all MP treatments resulted in a significantly shorter generation time, no significant differences were detected among the three concentrations (*p* > 0.05). Furthermore, individuals from the 20 mg L^−1^ treatment showed the lowest values (4.69 days), which could indicate a stress-induced acceleration of reproduction.

Growth rates of *M. macrocopa* ranged from 0.55 to 0.87 d^−1^. Statistical analysis revealed no significant differences between treatments (*p* > 0.05). The mean growth rates were 0.72 ± 0.05 d^−1^ in the control group, 0.68 ± 0.11 d^−1^ at 5 mg L^−1^, 0.77 ± 0.05 d^−1^ at 10 mg L^−1^, and 0.75 ± 0.07 d^−1^ at 20 mg L^−1^ (Figure 7). These results suggest that increase rates remained stable over time, implying a combination of reproductive strategies, at least in the F_0_ generation. Further studies are required to investigate these mechanisms and their possible impact on long-term population dynamics.

## 4. Discussion

Some authors have reported that microplastics in freshwater environments commonly range between 20 and 100 μm [32] and that cladocerans can ingest particles between 23 and 264 μm [33]. Similarly, it has been shown that *M. macrocopa* can grow and reproduce when feeding on organic particles between 35 and 40 μm [34], which supports the findings presented here.

Our results indicate that the effects of ABS secondary microplastics (10–150 µm) on the demographic variables of *Moina macrocopa* are not necessarily concentration dependent. Even at the lowest concentration tested, we observed detrimental effects on cladocerans comparable to those at medium and high concentrations (10 and 20 mg L^−1^). We suggest that the observed decrease in fitness may be due to intestinal obstruction and reduced efficiency of food intake. Both digestive obstruction and external adhesion impair cladoceran fitness, leading to abnormal movement and reduced feeding efficiency [35,36]. Consistent with this, our study found an 85% reduction in green algal consumption. Similarly, Reyes-Santillán et al. [37] reported that *Daphnia pulex* exhibited a 50% decrease in ingestion rate and altered heart rate when exposed to 30 µm plastic microspheres at 40 mg L^−1^. Even when plastic particles are not ingested due to their size, their mere presence can increase energy expenditure by interfering with algal consumption and requiring additional energy for the removal of clogged plastic particles [23]. De Felice et al. [38] reported that microplastics induced behavioral changes in terms of swimming activity, phototactic behavior, and reproduction.

Ingestion has been shown to be a critical factor in determining the negative effects of microplastics and represents the first indication of their mechanistic impact. *Moina macrocopa* has a non-selective filter-feeding behavior, which favors the ingestion of microplastics along with organic particles [39]. Consequently, microplastics lodge between the thoracic appendages and clog the gut. Although not measured, a personal observation we noted is that cladocerans frequently used their postadomen, whose function is cleaning, suggesting stress caused by the presence of particles. This is in agreement with the findings of Rehse et al. [39] and Frydkjær et al. [24], who reported that secondary microplastics, due to their irregular morphology, caused erratic movements.

Mortality in *M. macrocopa* increased, even at the lowest microplastic concentration, which is consistent with the findings by Castro et al. [40] for *D. similis*. The authors attributed this to appendage damage and reduced feeding. Similarly, Ogonowski et al. [41] reported increased mortality in *D. magna* due to lower algal consumption. Diet appears to play a critical role, as Tang et al. [42] found an increased expression of arginine kinase, an enzyme involved in energy production, in *D. magna* exposed to microplastics, suggesting an attempt to compensate for food limitation.

Fecundity and net reproduction were significantly reduced to 45% in the presence of ABS-MPs versus the control, contrary to what De Felice et al. [38] reported. They showed an increase in offspring production in *D. magna* exposed to primary microplastics. This discrepancy may be due to differences in microplastic type. Castro et al. [40] and Ziajahromi et al. [43] observed reduced fecundity in the cladoceran *Ceriodaphnia dubia* and *D. similis* cladocerans exposed to secondary microplastics, associated with the expended energy during swimming. Given the high energy demands of reproduction in cladocerans [44], gut blockage likely impaired nutrition, further reducing fecundity.

Interestingly, microplastics did not alter the timing of the first reproductive event in *M. macrocopa*, as both control and exposed individuals released their first offspring on the fourth day. This contrasts with De Felice et al. [38], who reported earlier reproduction in *D. magna* exposed to microplastics, possibly as a survival strategy. In our study, both fecundity and survival declined at the lowest microplastic concentration, suggesting that *M. macrocopa* did not prioritize reproduction over survival.

Cladoceran survival depends on whether they are exposed to primary or secondary microplastics. While smooth microbeads accumulate in the digestive tract without causing significant mortality [22,41], secondary microplastics with irregular morphology cause internal and external injury. Frydkjær et al. [24] found that cladocerans could rapidly remove smooth microspheres but retained those with irregular shapes, which compromised their survival. A similar pattern was observed in our experiments, where the life expectancy was reduced by approximately 35% in the presence of plastic materials.

## 5. Conclusions

In this study, the effects of three different concentrations of ABS-MPs on the life table of the freshwater cladoceran *M. macrocopa* were evaluated. It was observed that even the lowest concentration tested had significantly negative consequences on the life table of *M. macrocopa*, decreasing survival, life expectancy, and fecundity and increasing mortality. The interaction of zooplankton with the ABS particles obstructed their digestive system, causing immobilization and impaired swimming, which led to deficient feeding and declining nutrition.

## Figures and Tables

**Figure 1 biology-14-00555-f001:**
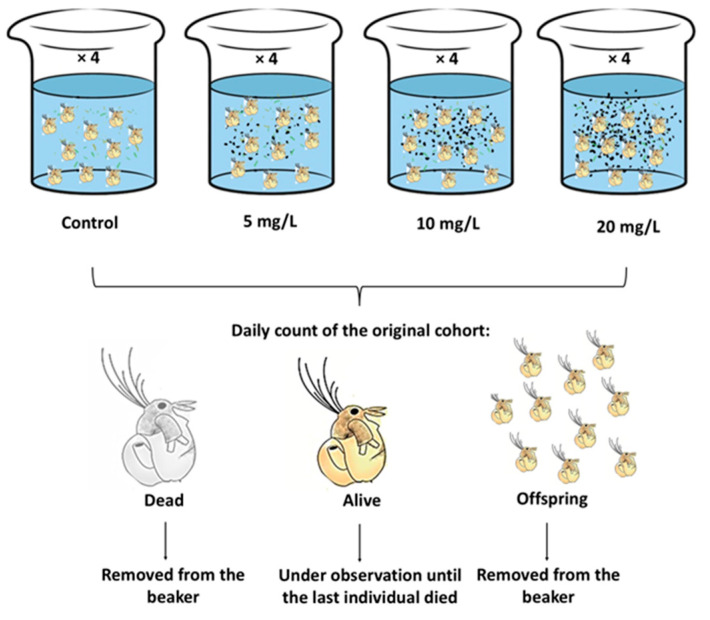
Experimental design.

**Figure 2 biology-14-00555-f002:**
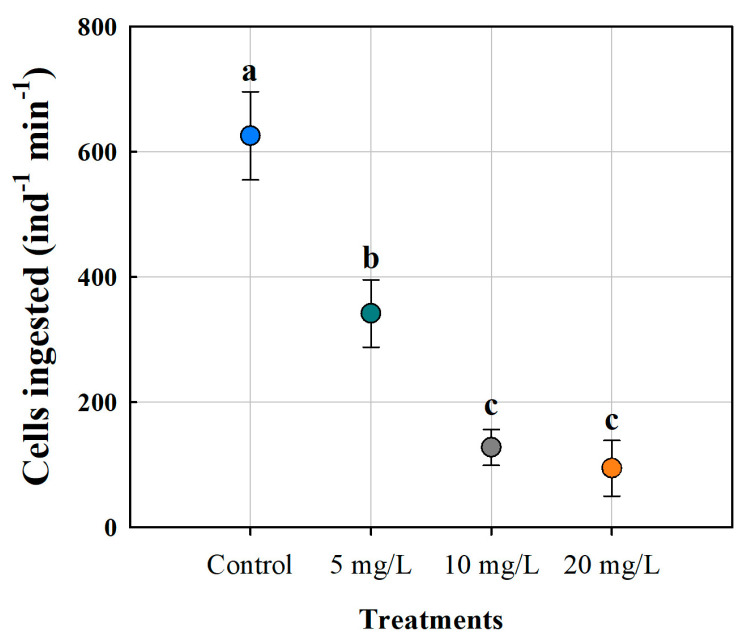
Ingestion of *C. vulgaris* by the cladoceran *Moina macrocopa americana* to ABS-MPs. The dot bars represent the mean and standard error of four replicates. Each color represents a distinct treatment. Different letters indicate a significant difference (*p* < 0.05, Tukey’s test).

**Figure 3 biology-14-00555-f003:**
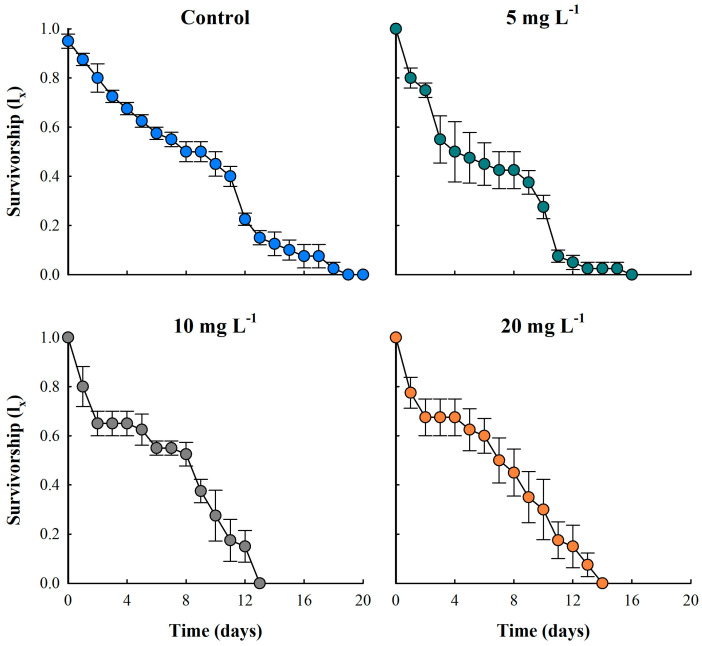
Survival of the cladoceran *Moina macrocopa* exposed to different concentrations of ABS secondary microplastics (5, 10, and 20 mg L^−1^) and control. Values are expressed as means ± standard error based on four replicates. Each color represents a distinct treatment: blue for control, green for 5 mg L^−1^, gray for 10 mg L^−1^, and orange for 20 mg L^−1^.

**Figure 4 biology-14-00555-f004:**
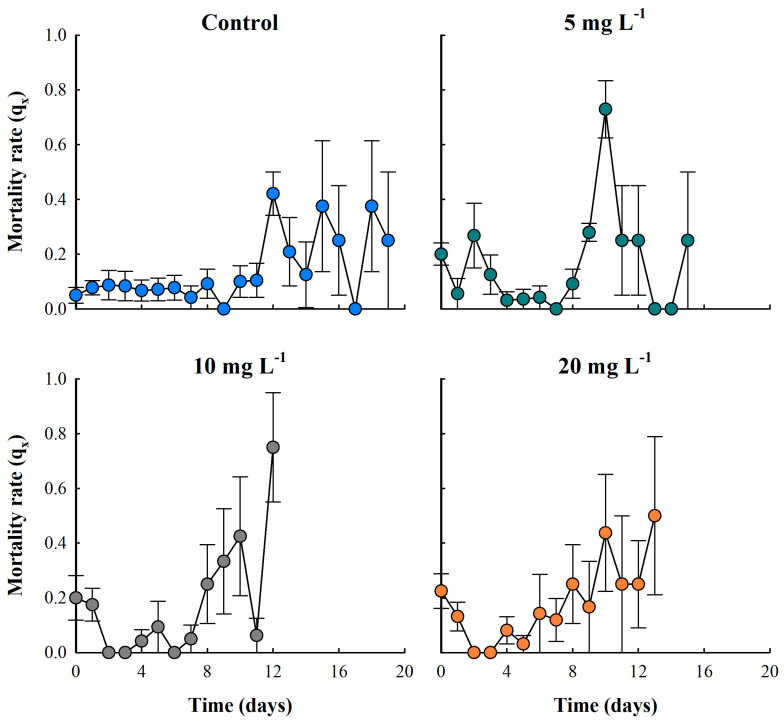
Mortality rate of the cladoceran *Moina macrocopa* exposed to different concentrations of ABS secondary microplastics (5, 10, and 20 mg L^−1^) and control. Values are expressed as means ± standard error based on four replicates. Each color represents a distinct treatment: blue for control, green for 5 mg L^−1^, gray for 10 mg L^−1^, and orange for 20 mg L^−1^.

**Figure 5 biology-14-00555-f005:**
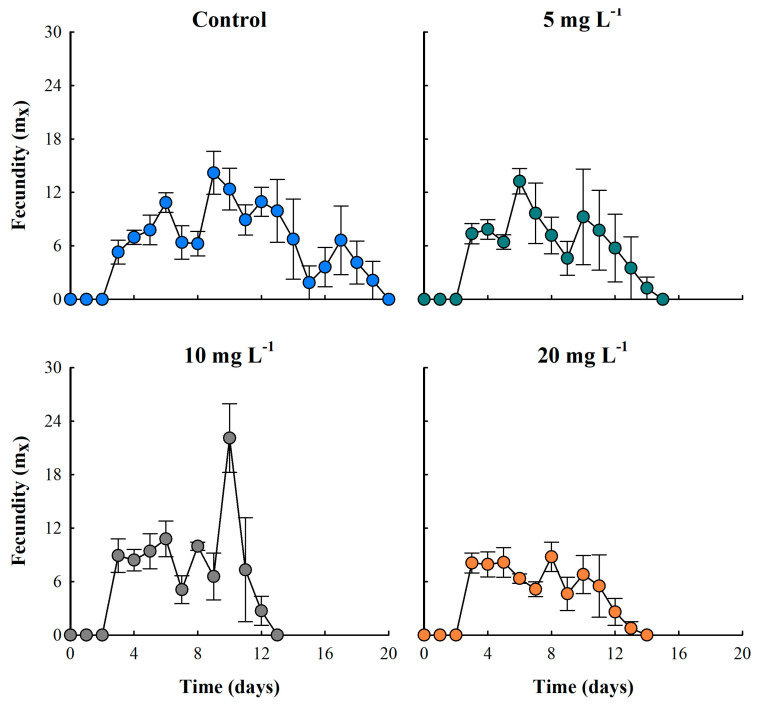
Fecundity (number of neonates per female) of the cladoceran *Moina macrocopa* exposed to different concentrations of ABS secondary microplastics (5, 10, and 20 mg L^−1^). Values are expressed as means ± standard error based on four replicates. Each color represents a distinct treatment: blue for control, green for 5 mg L^−1^, gray for 10 mg L^−1^, and orange for 20 mg L^−1^.

**Figure 6 biology-14-00555-f006:**
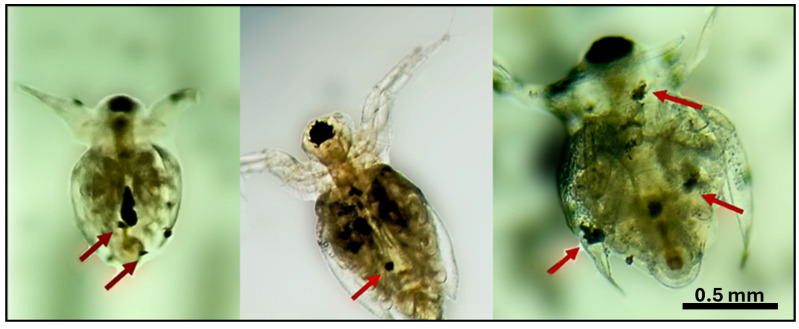
Red arrows highlight secondary microplastics blocking the appendages and gut of *Moina macrocopa*, with additional particles adhering to the carapace.

**Figure 7 biology-14-00555-f007:**
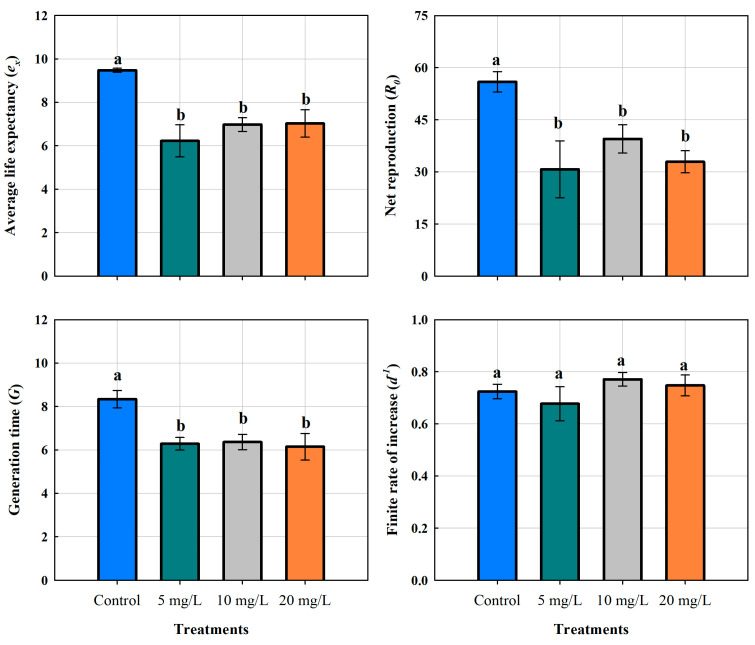
Life expectancy of the cladoceran *Moina macrocopa*, generation time, net reproduction, and finite rate of increase when exposed to different concentrations of ABS secondary microplastics. Values are expressed as means ± standard error based on four replicates. Each color represents a distinct treatment: blue for control, green for 5 mg L^−1^, gray for 10 mg L^−1^, and orange for 20 mg L^−1^. Different letters indicate a significant difference (*p* < 0.05, Tukey test).

**Table 1 biology-14-00555-t001:** One-way analysis of variance applied to ingestion rates of *Moina macrocopa* exposed to ABS-MPs. The table provides information on degrees of freedom (DFs), the sum of squares (SS), mean squared (MS), Fisher’s test statistic (F), and the corresponding *p*-values.

Source of Variation	DFs	SS	MS	F	*p*
**Between Groups**	3	179,587.383	59,862.461	22.398	<0.001
**Residual**	12	32,071.468	2672.622		
**Total**	15	211,658.851			

**Table 2 biology-14-00555-t002:** One-way analysis of variance (ANOVA) applied to the demographic variables of *Moina macrocopa* exposed to three concentrations of ABS microplastics (5, 10, and 20 mg L^−1^). The reported statistical parameters include degrees of freedom (DFs), sum of squares (SS), mean square (MS), Fisher’s test statistic (F), and the corresponding *p*-values.

Source of Variation	DFs	SS	MS	F	*p*
**Average lifespan**					
Between groups	3	25.007	8.336	10.474	0.001
Residual	12	9.550	0.796		
Total	15	34.557			
**Net reproductive rate**					
Between groups	3	2114.262	704.754	10.429	0.001
Residual	12	810.915	67.576		
Total	15	2925.177			
**Generation time**					
Between groups	3	12.974	4.325	5.826	0.011
Residual	12	8.908	0.742		
Total	15	21.882			
**Population growth rate**					
Between groups	3	0.0191	0.00638	0.872	0.483
Residual	12	0.0879	0.00732		
Total	15	0.107			

## Data Availability

All the data produced or examined in this study are provided within this published article.

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
