# Peer review of "Effect of Acrylonitrile Butadiene Styrene (ABS) Secondary Microplastics on the Demography of Moina macrocopa (Cladocera)"

_biology, 2025, doi:10.3390/biology14050555_

Round 1
Reviewer 1 Report
Comments and Suggestions for Authors
This study investigated the impact of secondary microplastics from ABS on the population dynamics of freshwater cladoceran Moina macrocopa. The topic is of significant ecological importance, with particular attention paid to the potential risks of non-standard plastic types (ABS) in freshwater ecosystems. The experimental design is rigorous, and the data presentation is clear. However, the layouts of figures and data statistics need further improvements. The detail suggestions listed as follows:
- Authors mentioned some parameters were found to be non-significant in your study, but table 1 only listed P value between groups, a more detail statistics of all indicators between groups, i.e. survival, ,mortality, life expenctancy, fecundity and etc.
- You centralized you figure together, all of which are far from their relevant content, such as you mentioned fig 2 in line 186, but I can only find in line 293, you should reorganise them with content in context.
- Delete g in line 98 before 1.9 g.
Author Response
Effect of Acrylonitrile Butadiene Styrene (ABS) secondary microplastics on the demography of Moina macrocopa (Cladocera)
|
1. Summary |
|
|
|
Thank you very much for taking the time to review our manuscript. Please find the detailed responses below and the corresponding revisions/corrections highlighted/in track changes in the re-submitted files. |
||
|
Reviewer 1 Comments 1: Authors mentioned some parameters were found to be non-significant in your study, but table 1 only listed P value between groups, a more detail statistics of all indicators between groups, i.e. survival, mortality, life expenctancy, fecundity and etc]
|
||
|
Response 1: Thank you for your observation. In addition to the one-way ANOVA, we performed a Tukey's post hoc analysis to confirm significant differences between treatments and modified Figure 2 to highlight the differences.
|
||
|
Comments 2: You centralized you figure together, all of which are far from their relevant content, such as you mentioned fig 2 in line 186, but I can only find in line 293, you should reorganize them with content in context. Response 2: We agree with the comment; however, when the editorial team organized the information, the figure was placed immediately after its mention in the text. This can be verified in the preprint. |
||
|
Comments 3: Delete g in line 98 before 1.9 g
|
||
|
Response 3: Deleted |
||

Reviewer 2 Report
Comments and Suggestions for Authors
The paper aim the effects of microplastic ingestion on survival, mortality, life expectancy, fecundity, and feeding rates of Moina macrocopa. The topic is highly relevant, especially considering the impacts of microplastics on aquatic ecosystems, which reinforces the importance of the study. Overall, the study is well written and suitable for publication in Biology. The introduction is clear and addresses the most important aspects of the study, the methods used are appropriate, and the results and discussions are well presented and well-founded. The conclusion also reflects the main findings in a coherent manner.
My main highlights are:
- Statistical analyses: I suggest that the authors include information on the normality and homogeneity tests performed.
- Presentation of tables: The tables accompanying the results seem to summarize the information presented in the figures but could be organized more clearly. It might be interesting to transfer them to the supplementary material, including the values ​​obtained in each experimental group. This would make the data easier to understand.
- Scale bar: I suggest adding a scale bar to Figure 5.
In short, I consider the paper to be well developed and relevant. My overall recommendation is "minor revisions", as the necessary changes are specific and aim to improve the clarity and organization of the results.
Author Response
Effect of Acrylonitrile Butadiene Styrene (ABS) secondary microplastics on the demography of Moina macrocopa (Cladocera)
|
1. Summary |
|
|
|
Thank you very much for taking the time to review our manuscript. Please find the detailed responses below and the corresponding revisions/corrections highlighted/in track changes in the re-submitted files. |
||
Reviewer 2
Comments 1: Statistical analyses: I suggest that the authors include information on the normality and homogeneity tests performed.
Response 1: Thank you for your comment, we have now added information in the Statistical Analysis section. SigmaPlot uses the Kolmogorov-Smirnov test to test for a normally distributed population.
Comments 2: Presentation of tables: The tables accompanying the results seem to summarize the information presented in the figures but could be organized more clearly. It might be interesting to transfer them to the supplementary material, including the values ​​obtained in each experimental group. This would make the data easier to understand.
Response 2: We appreciate your feedback on this matter. We believe that the tables provide valuable supplementary information to the graphs presented. While the graphs illustrate the overarching trends, the ANOVA graphs enhance our understanding of the differences observed.
Comments 3: I suggest adding a scale bar to Figure 5.
Response 3: Added
